# Winter City Urbanism: Enabling All Year Connectivity for Soft Mobility

**DOI:** 10.3390/ijerph16101820

**Published:** 2019-05-22

**Authors:** David Chapman, Kristina L. Nilsson, Agatino Rizzo, Agneta Larsson

**Affiliations:** 1Architecture Group, Luleå University of Technology, 971 87 Luleå, Sweden; kristina.l.nilsson@ltu.se (K.L.N.); agatino.rizzo@ltu.se (A.R.); 2Health Sciences, Luleå University of Technology, 971 87 Luleå, Sweden; agneta.larsson@ltu.se

**Keywords:** soft mobility, walkable environment, physical activity, health outcomes, active living

## Abstract

This study explores connectivity for soft mobility in the winter season. Working with residents from the sub-arctic city of Luleå, Sweden, the research examines how the interaction between the built environment and winter season affects people’s use of the outdoor environment. The research questions for this study are (1) How do residents perceive the effects of winter on an areas spatial structure and pattern of streets and pathways? and (2) What enablers and barriers impact resident soft mobility choices and use of the public realm in winter? Methods used were mental mapping and photo elicitation exercises. These were used to gain a better understanding of people’s perception of soft mobility in winter. The results were analysed to identify how soft mobility is influenced by the winter season. The discussion highlights that at the neighbourhood scale, residents perceive that the winter alters an areas spatial structure and pattern of streets and pathways. It was also seen to reduce ease of understanding of the public realm and townscape. In conclusion, it is argued that new and re-tooled town planning strategies, such as extending blue/ green infrastructure planning to include white space could help better enable all year outdoor activity in winter cities.

## 1. Introduction

All over the world, the form of the built environment plays a key role in enabling urban outdoor activities such as soft mobility. The public realm can make it more attractive for people to be mobile outdoors and to participate in public life or it can put people off venturing outside.

For the purpose of this study, soft mobility is defined as human-powered, non-motorized ways of getting around, that have a relatively little impact on the environment, while connectivity in the built environment is defined as the degree to which a place and its parts are connected to each other [1]. Together connectivity for soft mobility, which collectively can be defined as how the built environment influences soft mobility choices in the form of walking and cycling, has been linked to a range of town planning agendas. It was been linked to reducing building and transport pollution [2,3] and more efficient use of land [4]. It has been associated with discussions around energy efficient modes of transport and reduced car usage [5]. It has been linked to the human wellbeing agenda by helping facilitate physical activity [6,7]. Today, creating active built environments that enable regular, all year outdoor soft mobility responds to the United Nation’s Sustainable Development Goals and the World Health Organization’s Global action plan on physical activity 2018–2030 (2018).

While there are many arguments for why connectivity for soft mobility is an essential part of the urban design of settlements, there is currently little knowledge of how it is affected by seasonal climate variation. As such, the aim of this study is to explore how the interaction between the built environment and winter season create barriers and enablers to connectivity for soft mobility. 

The main research questions were (1) how do residents perceive the effects of winter on an areas spatial structure and pattern of streets and pathways? and (2) what enablers and barriers impact resident soft mobility choices and use of the public realm in winter?

### 1.1. Climate-Sensitive Urban Design

Settlements are usually discussed as being in a state of constant evolution or change rather than being static or, finished [8]. Equally, for those involved in the design and planning of the built environment, such changes are most commonly associated with physical, social, cultural or economic conditions, rarely are the changes created by seasonal climate investigated.

For urban design connectivity for soft mobility is a common focus, whether it is described as walking and cycling, connectivity, permeability, integration or just merely, ease of movement [9,10,11,12,13]. For most designers and planners, soft mobility is a critical dimension of urban design that has been subject to numerous publications [14]. For a detailed review, see Stephen Marshall’s book, *Streets & Patterns* [15]. However, within these studies the implications of seasonal climate on connectivity for soft mobility is less well understood. This is important today because settlements that experience significant seasonal climate variation are challenged with enabling all year outdoor soft mobility as part of a range of town planning and health policy agendas.

For such settlements, seasonal climate variation can significantly complicate the design of urban form and public realm. Here it is often seen that the interaction between the built environment and seasonal climate variation can physically affect the public realm and the levels of connectivity for soft mobility an area can afford.

### 1.2. Winter City Urban Design

Winter settlements are usually places that experience high degrees of seasonal climate variation, temperatures commonly below zero, precipitation that is mainly snow, and limited hours of sunshine and daylight [16]. In such settlements, winter conditions and covers of snow and ice are a regular part of the built environment and can be in place for between 4–6 months of the year.

Research into the design of winter cities has been ongoing since around the middle of the twentieth century and reached a peak in the 1980s and 1990s [17]. Notable advocates of climate-sensitive urban design for winter settlements were the Swedish-English architect Ralph Erskine and the Canadian planner Norman Pressman. Erskine’s 1959 ‘A Grammar for High Latitudes’ [18] provides an early example of climate-sensitive design guidelines for winter communities. Here, Erskine highlights the importance of considering the cold, heat, snow, frost, light, wind, vegetation and the microclimate in the design process [18].

Pressman published numerous books and articles on winter cities between 1986 and 2004. While his works looked at a range of different ways of making winter cities more liveable, his work settled and focused on three climate-sensitive design principles for winter cities. Here his microclimatic design ideas focused on maximising solar access, minimising the negative effects of wind and managing snowfall and gathering [19,20,21,22]. 

The work of Erskine, Pressman and others including the Winter Cities institute (wintercities.com) is influential in improving our understanding of winter city urban design and a detailed state of the art of this work can be found in *Updating Winter: The Importance of Climate-Sensitive Urban Design for Winter Settlements*, Arctic Yearbook, 2018 [17]. However, while these works cover many dimensions of winter urbanism, they do not address how people perceive connectivity for soft mobility in winter.

## 2. Materials and Methods 

The study focused on a single case study neighborhood where it investigates resident’s perceptions of the urban structure and barriers and enablers to soft mobility in streets and spaces in the winter season. A sequential mixed methods design was used [23] combining two qualitative participatory methods; mental mapping and photo-elicitation method. This design was chosen as the combination of the two methods provides a deeper understanding of the topic. 

As both methods enables residents to record and reflect their neighborhood, the results from each method can be brought together for a more detailed analysis about how the interaction between the built environment and winter season creates barriers and enablers to connectivity for soft mobility. The results from the mental mapping focuses on the urban scale. Here, the findings will provide an insight into how residents perceive the urban structure of their neighbourhood environment and surroundings in winter conditions, in comparison to summer conditions. The results from the photo-elicitation exercise will deepen these results and explore how the public realm (the streets and spaces) of the neighbourhood are altered by the winter season.

### 2.1. Case Study

The city of Luleå, Sweden is located at 66.5622° N (latitude) just below the Arctic Circle (Figure 1). It is identified in the Köppen–Geiger Climate Classification system as sub-arctic. In the summer temperatures can reach +30 degrees and the sun does not set for significant periods. During the winter, temperatures can reach −30 with minimal daylight hours. The sea freezes annually for 6–7 months. The selected neighbourhood for the research was Mjölkudden, Luleå. Mjölkudden is a mixed-use neighbourhood and has a residential population of 3491. The average age is 46 years and there is an almost equal male to female ratio [24]. The area contains a variety of functions including a healthcare centre, pharmacy, dentist, church, supermarket and leisure facilities. Housing accommodation is both single and multi-family houses.

### 2.2. Data Collection

The study focused on gathering in-depth data from a group of residents from the case study neighbourhood. Study participants were recruited from staff at the university and the residential home care services unit. An invitation letter containing information about the content and date of workshops and inclusion criteria for the study was provided by email. Criteria for inclusion was being a resident in Mjölkudden and regularly moving by foot or bicycle in the area in wintertime. The ambition was to include a variety of persons with different capacities and ages, to reflect the demographics of residents in the location. In total, 15 residents of the case study neighborhood signed up to take part in the study and of these nine were male and six were female. The range of participants was in active adult age from early twenties to late fifties. 

Two sequential workshop sessions led by the first author were conducted in a meeting room at the university. The first meeting focused on introducing the research purpose and the methodology to the participants and the preparation of mental maps. The second meeting focused on the photo elicitation method and semi-structured group discussions. This sequence was repeated twice to offer suitable dates and form groups of 5–9 residents to support creative reflection and dialogue.

The study was performed in accordance with the ethical principles of the Helsinki Declaration, and informed consent was obtained from each participant.

#### 2.2.1. Mental Mapping

In the first workshop, each participant was asked to prepare a number of mental maps of the neighbourhood. To ensure that all participants had a clear understanding of what mental maps were and their use in understanding urban form, Kevin Lynch’s methods for urban analysis and example mental maps were discussed at the start of the meeting [25,26]. Each participant was asked to draw one winter and one summer mental map for each of the following questions:
Draw a quick sketch map of Mjölkudden showing the most interesting and important features, and giving a stranger enough knowledge to move about without too much difficulty and avoid major barriers.Make a similar sketch of the route and events along a typical trip (using soft mobility) from Mjölkudden to the City Centre.

After this meeting, the main author analysed all of the maps to identify recurring features and patterns. Once, these features and patterns were identified, this information was synthesised into two plan based images of the neighbourhood and surroundings. One map was drawn for the winter neighbourhood and one for summer. All maps were drawn to the same scale and use Lynch’s standard notation of Path, Edge, Node, District, and Landmarks [25,26].

#### 2.2.2. Photo-Elicitation

At the end of the first meeting, the photo-elicitation method was introduced. For this second workshop, participants were asked to photograph physical aspects of the neighbourhood that they perceived as facilitators or barriers to connectivity for soft mobility in winter. Participants were given two weeks to take these pictures and at the end of this period, they were asked to email their ten most relevant pictures to the principal author. Here participants were left to self-select their preferred images. 

At the start of the second meeting, each participant received a full set of A4 prints of his or her photographs. Participants were then asked to write a brief description of the image on the back of each photograph and then rank them in order from most significant barrier to greatest enabler. 

This prioritisation was used to encourage participants to identify issues of the most significant importance. Once everyone had ordered their pictures, he or she were asked to lay them out of tables in sequence. The facilitator then led an open discussion around each participant’s ordered set of photographs. The dialogue intended to allow for a free exchange of opinions and facilitate critical reflection. Each participant explained their reasoning behind taking each photograph and the message each image was trying to convey. The dialogue was audiotaped and later analysed by the main author in order to identify principal content in terms of emerging issues, recurring themes, and representative quotations.

## 3. Results

### 3.1. Mental Mapping—Urban Scale

The two plan-based images for winter and summer respectively, both highlighted the local center, which houses retail and community facilities, as a node and the church as a landmark. Equally, winter and summer maps illustrated a number of distinct residential areas or districts. Maps also showed the main strategic connections to the city centre and the university area but often lacked the smaller routes that can be found in the neighbourhood. Where the winter and summer maps differed was in the description of the outdoor environment and the landscape. Summer illustration show the green and blue spaces of the neighborhood as separate identified areas, while the winter map shows these areas to merge into one and form one overall white space of snow and ice. Winter illustration also highlighted a range of soft mobility options on the frozen sea. While, the ‘ice-road’, which is a formal winter route connecting the city’s northern and southern harbours via a route along the peninsula, featured prominently, individual participant maps also showed that areas outside the formal route were also used for soft mobility. Here the maps highlighted that in general, the ice parts of this ‘white space’ enabled soft mobility, whether set out as a formal route or not (Figure 2).

### 3.2. Photo-Elicitation—Public Realm

Participant photographs of the neighbourhood deepened the results from the mental mapping and showed how the winter season changed the local network of streets and pathways. Here discussions focused on terrain conditions and how the public realm and townscape changed in the winter season. Participants highlighted that build-ups of snow and ice reduced the usable area of the public realm, altered the local network of pathways in the winter season and changed the townscape of the neighbourhood. At a structural level, participants stated that build-ups changed the neighbourhood’s network of pathways and townscape. They highlighted that in the winter, ‘walkways disappear and then you need to cycle on the roads’. They also stated that ‘summer traffic management solutions coupled with snow piling blocked many pathways’ in winter. This was seen to have two impacts for connectivity for soft mobility in the winter season. The first being that the number of routes available for soft mobility was less in winter than summer. The second being that in winter, remaining routes were more likely to be on bigger vehicular routes. Both were seen as detractors for walking and cycling (Figure 3).

Similarly, the winter season were seen to change the townscape of the neighbourhood and the neighbourhoods visual appearance. Here it was suggested that the look of the neighbourhood ‘changes every week and it always looks different depending on the snow’. While this was not discussed as a barrier to soft mobility, it was seen to create safety issues for pedestrians and cyclists as it often reduced route visibility and masked vehicle noise (Figure 4).

The photo elicitation exercise also highlighted that the winter season made it more difficult for people to identify the different elements that make up the public realm. Here the winter season was described as having a ‘whiteout effect that made it difficult to understand the area’. For example, participants stated that it was hard to understand where the pedestrian and cycleway were or the extents of the roadway. Participants saw this whiteout effect as a detractor for soft mobility as it made it unclear which modes of transport had priority. On top of this participants highlighted, that for soft mobility, relatively wide pavements in summer became narrower in the winter season. Together the whiteout effect and the reduction in the usable area of the public realm were seen as a significant barriers to soft mobility and potential sources of conflict between different modes of transport (Figure 5).

It was also discussed that, ‘if it was very cold, the conditions are OK to go with the bike, the problem is around zero temperatures’. Within reason, coldness was not seen as a barrier to soft mobility. However, many saw that main barriers to soft mobility occurred when temperatures were around zero degrees Celsius. Around this temperature, mixed conditions of snow, ice, slush and water were likely to build on the ground. Participants described this ‘as the worst’ and this was seen by all to be unpleasant and a significant barrier to soft mobility. Comparatively, participants highlighted few enablers to soft mobility in winter. The removal of snow and the grading of streets and pavements were seen as the major enabler to soft mobility. For cycling, the provision of high-quality cycle storage areas was also highlighted as a significant enabler.

## 4. Discussion

The study aimed to explore how the interaction between the built environment and winter season creates barriers and enablers to connectivity for soft mobility. 

The mental mapping exercise showed that broadly speaking people’s image of the ‘neighbourhood’, regards nodes, landmarks and districts was similar in the winter and summer season. However, they also showed that peoples image of the outdoor environment was very different depending on the season. Here the neighbourhood maps showed that the summer and autumn blue and green areas of the neighbourhood became white with the winter season. In winter maps, the frozen sea was seen to support a range of soft mobility choices including walking, cycling, skiing and skating. Mapping also showed that both prepared and unprepared ice can both connect communities and facilitate soft mobility. These outcomes suggest to enable all year outdoor activity, and the health benefits this can bring, such communities need to plan for soft mobility when the outdoor environment is both in its blue and green form and when it becomes white due to the winter season [17,27]. 

The manner in which participants illustrated the ice-road was significant (Figure 6). In drawings, the route was shown in more detail than other routes and was often annotated. This suggested that the route established a strong image in people’s mind and gave a good indication that the route was perceived as an important part of the winter image of the city. However, participant maps clearly showed that the ‘ice-road’ starting some distance away from the shore. This suggested that participants did not perceive the ‘ice-road’ as being connected into the neighourhood’s land-based networks of streets and pathways. Here, to enable outdoor soft mobility in winter, planners should consider how ephemeral winter routes connect in to the settlements permanent networks of streets and routes.

Results from the photo-elicitation highlighted that the interaction between the built environment and winter season created a range of affects to soft mobility. The photographs highlighted that different terrain conditions, such as ice and slush were seen as barriers to soft mobility [28,29]. Importantly, the discussions also established that the winter season was seen to be significant enough to alter the urban structure of the neighbourhood, its public realm and the townscape, and create safety issues [30,31] (Figure 7). Interesting, however, the negative impact by ice and snow on outdoor activity that is commonly reported [32], is challenged by the participants’ positive view of the ice road. Which implies that icy and snowy terrain conditions can have different qualities (depending on design and maintenance) [27]. The results also showed that physical barriers to soft mobility were compounded by the ‘white-out’ effect created by the interaction between the built environment and winter season. Here the terrain covers of ice, snow, or slush and the grading of the public realm was seen to reduce the public realm surface to one large undefined area.

### 4.1. Limitations of the Study

To obtain the required results of the study we used a mixed method design with qualitative participatory methods. An advantage with this approach is that it provided a deeper understanding about how residents in the case study location perceive barriers and enablers to soft mobility in winter. The inclusion of a variety of participants suggests that they represent the voices of residents in this location. Also the case study location can be seen as representing a common neighborhood area in a northern Nordic location. However, there are number of factors that may have influenced residents’ willingness to participate in this study, which imply that they may differ in some regard from other residents. For example, those perceiving a higher commitment and outdoor activity might have been more inclined to participate. However, increased knowledge of the residents’ mental models as basis for individuals’ decision making for soft mobility in winter is important, as it reflects the daily use of the network of streets and walkways for mobility. Hereby the results also sheds lights on what is successful or lacking in the planning and maintenance in a common neighborhood area in northern cities in this point of time. The knowledge based on residents perceptions and synthesised by the main author, can form the basis for following workshops with city planner and policy makers. It can also guide the design of an implementation study where the neighborhood is a testbed for experiments.

### 4.2. Practical Implications and Suggestions for Further Research

As the results highlight that people perceive the neighbourhood’s structure and appearance to be changed by the winter season, those involved in enabling connectivity for soft mobility should seek to understand these changes and design and plan for winter urban movement strategies. For soft mobility, such places, like their summer equivalents, should focus on creating a desired winter urban structure of connected streets and spaces, with an understandable townscape coupled with a high quality and useable winter public realm. Planners and designers should also focus on reducing ambiguity in the public realm that is created by the winter ‘white-out’ effect and seek design solutions that reduce confusion about the user priorities in an area. This could be achieved by strategies of elevating movement information above ground level, using snow and ice to define different circulation pathways or using projection and light to define space (Figure 8).

For policy-makers and practitioners, the results show three main outcomes regarding the interaction between the built environment and the winter season and the effect on connectivity for soft mobility. These interactions are seen to reduce the spatial structure of an area and the network of streets and pathways that are for soft mobility. It has also been seen to create ambiguity in the public realm and alter a places townscape. All of which have been identified by participants as barriers to soft mobility.

## 5. Conclusions

This study has shown that the winter season creates barriers to outdoor soft mobility in the built environment, which in turn limits the opportunities for the health benefits regular all year outdoor soft mobility can bring. It can be concluded that in winter settlements, focusing solely on the weather and climatic dimensions of the winter season is too limited an approach when designing and planning for these places. Instead, it is important to focus on how the interaction between the winter season and built environment alters the urban structure and public realm of the settlement. That is when blue and green spaces of the outdoor environment are free of snow and ice and, when they have the white cover of the winter season.

This could be done by bringing forward new types of planning strategies for winter settlements and/or re-tooling existing planning methods. For example, blue-green planning strategies can be extended to blue-green-white planning strategies. In winter cities, these strategies would address the structure, function and design of green and blue public areas, spaces, streets and paths when they become white due to snow and ice. These plans would seek to achieve an attractive built environment where transport by walking and biking is prioritised and inviting as an everyday activity throughout the year. These plans would focus equally on winter and summer connections and pathways for soft-mobility, formal vehicular infrastructure and public space maintenance and management. As the winter season is dark they would address the structure, function and design of lighting. At a technical level, these plans would address snow removal and storage [27]. The re-tooling of such strategies for both summer and winter would enable designers and planner to better envisage and design for how the public realm operates in both situations. Both, in turn, would help public policy enable higher levels of outdoor soft mobility in the winter season and help bring the individual health benefits that come from outdoor activity.

## Figures and Tables

**Figure 1 ijerph-16-01820-f001:**
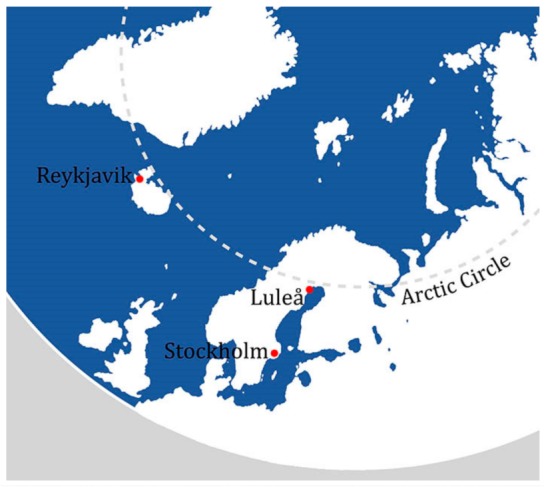
Location map for Luleå, Sweden.

**Figure 2 ijerph-16-01820-f002:**
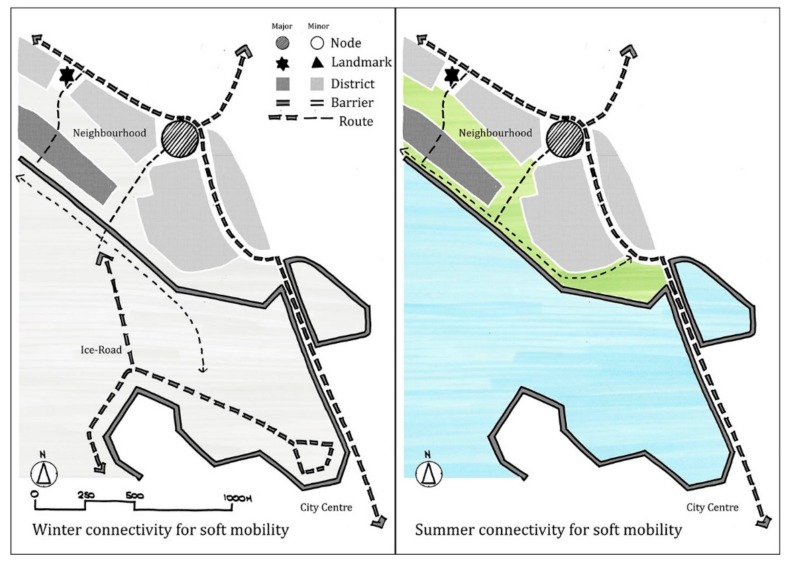
Plan based images compiled from participant mental maps show how the neighbourhood’s connectivity options for soft mobility change in the winter (left) and summer (right).

**Figure 3 ijerph-16-01820-f003:**
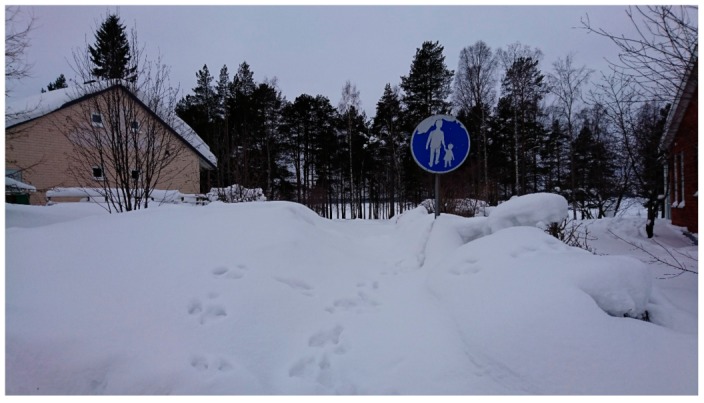
A participant’s photograph shows how walkways can ‘disappear’ in the winter season.

**Figure 4 ijerph-16-01820-f004:**
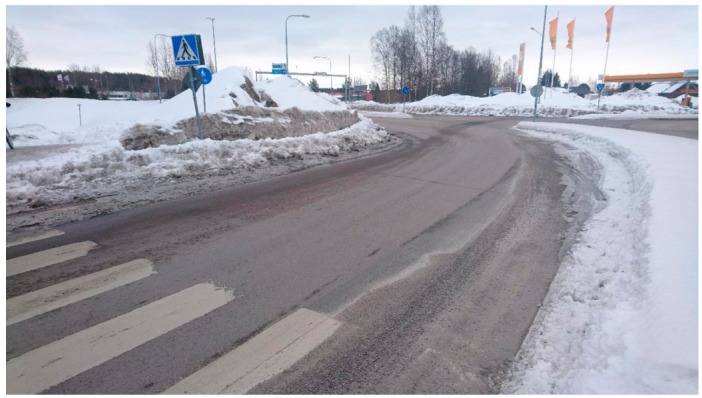
A participant’s photograph shows how walkways can ‘disappear’ in the winter season.

**Figure 5 ijerph-16-01820-f005:**
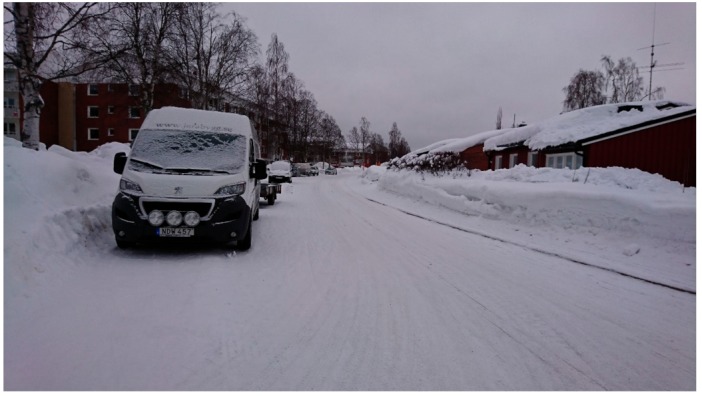
A participant’s photograph illustrates how the snow can ‘white-out’ the street and its features.

**Figure 6 ijerph-16-01820-f006:**
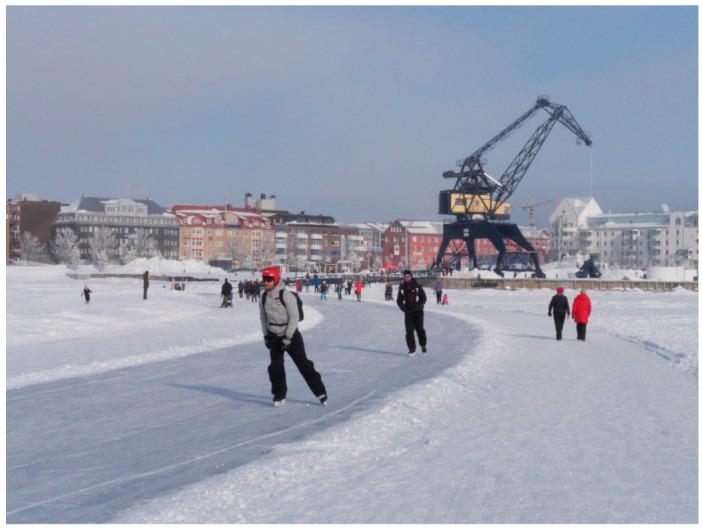
Photograph of Luleå’s ice road looking back toward the City’s southern harbour.

**Figure 7 ijerph-16-01820-f007:**
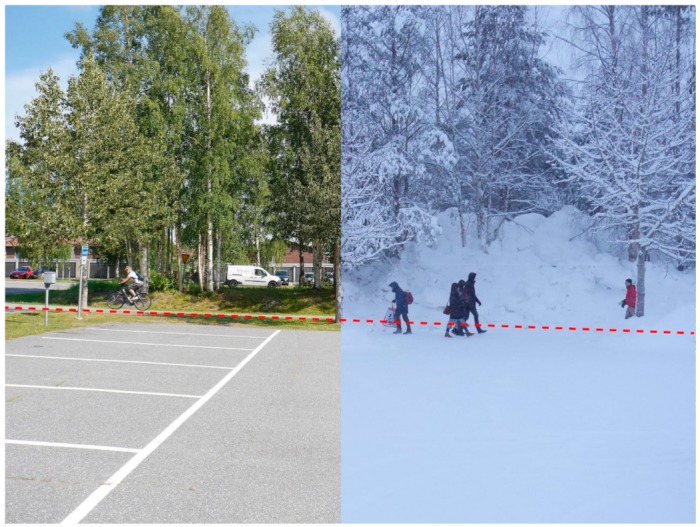
An illustration of how the image of the same area in summer (left) and winter (right) can change due to weather conditions (copyright: David Chapman).

**Figure 8 ijerph-16-01820-f008:**
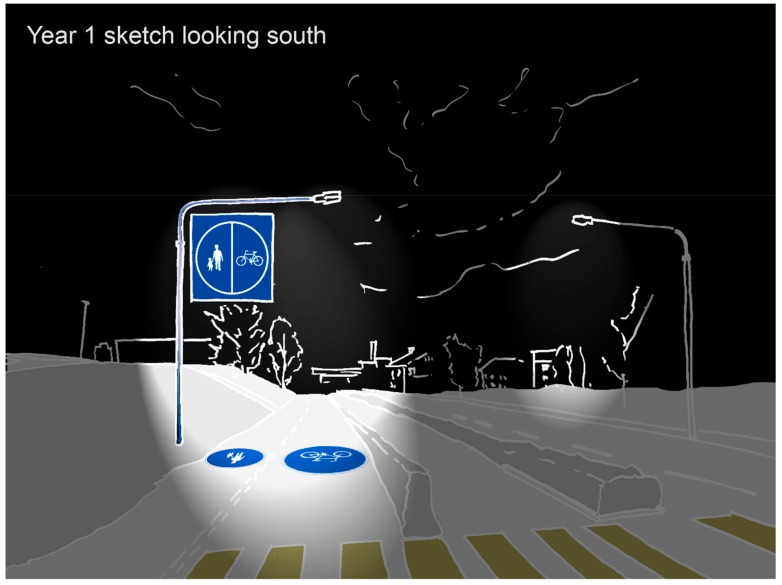
An illustration of seasonal winter cycle lanes that are being tested by Lulea University of Technology in Kiruna, Sweden. Here winter cycle lanes are marked out using projected information on to the snow and ice and separated from vehicular traffic by low snow walls (copyright: David Chapman).

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
