# Peer review of "Winter City Urbanism: Enabling All Year Connectivity for Soft Mobility"

_ijerph, 2019, doi:10.3390/ijerph16101820_

Round 1
Reviewer 1 Report
This manuscript examines how seasonal changes in the environment (especially winter) influence soft mobility and connectivity in a Winter City. The project employed mental mapping and Photovoice to elicit participants' perceptions of changes in mobility by season.Overall, I found the manuscript very brief and was left with many questions. These questions are provided as a way to potentially strengthen the work.
1) What is the relationship between the larger area (as portrayed in the mental maps), versus the neighbourhood? It was not clear to me whether the two methods were examining exactly the same area, or if the neighbourhood was embedded in the larger mental map region. Perhaps a map would help with this and could identify where Lulea is in Sweden, as well as where this particular neighbourhood is situation within the city. This context would be helpful for understanding the relationship between the two data collection methods.
2) Conceptually, I was not clear on whether the work focused on connectivity of the built environment, or soft mobility of the individual. This is an important distinction. Connectivity has to do with how environments can constrain or enable a particular behaviour, while soft mobility has to do with the individual (and sometimes group) decision about how to move around. Throughout the manuscript, the authors refer to "connectivity for soft mobility" and I was left wondering, exactly, what they meant by that term.
3) Over the past 5 years there has been a growing interest (again) in Winter Cities. I encourage the authors to access this newer work, some of which is referenced or available at WinterCities.com.
4) The authors also refer to participants' use of green and blue spaces. I don't know if it is directly applicable, but the authors might consider looking at Robin Kearns new book: Blue Space, Health and Wellbeing.
5) ON page 5 (lines 167-175) the findings here (no winter connectivity/mobility) seem to contradict the findings on page 4 (lines 148-161) that suggest that the ice road increases connectivity. I suspect that this is related to my first point - above - that there is some muddiness in the spatial extent of the two data collection methods.
6) The conclusions do seem somewhat self-evident. I, too, live in a Winter City. Certainly different planning strategies are required when considering how spaces will (or will not) be used in the different seasons. Could the authors, perhaps, provide more concrete examples of such proposed "re-tooling"? What would that look like? Because the reality of making such changes happen is complex. For example, cycling lanes or paths could be instituted (capital or infrastructure costs) but use throughout all four seasons required operating budgets to clear those paths of snow and ice. So such budgeting considerations must be part of any such 're-tooling'.
Finally, two issues.
First, I found the manuscript overall far too brief. I am not sure if this is due to the journal's word limit, or simply the assumption of the authors that the readers are from Sweden and understand the constraints and challenges of that particular milieu. I would like to see the findings and discussion sections expanded and more complexity and nuance provided.
Second, the manuscript needs a careful and extensive edit for english language. For example, even the title states "...enable all year around...". This is not the conventional use in English. It is generally referred to "all year" or (more casually) "year round". Throughout the manuscript there are incomplete sentences ("That is...") extensive use of passive voice, and some awkward phrasing throughout. A simple pass through an English-speaking editor would easily catch and fix these simple errors.
Author Response
Point 1: Overall, I found the manuscript very brief and was left with many questions.
Response 1: The paper has been extended by just under 1000 words to address all reviewer comments.
Point 2: What is the relationship between the larger area (as portrayed in the mental maps), versus the neighbourhood? It was not clear to me whether the two methods were examining exactly the same area, or if the neighbourhood was embedded in the larger mental map region. Perhaps a map would help with this and could identify where Lulea is in Sweden, as well as where this particular neighbourhood is situation within the city. This context would be helpful for understanding the relationship between the two data collection methods.
Response 2: One map has been added that identifies the location of LuleĂĄ in Sweden. Instead of adding a map of the study neighbourhood, parts of the material and methods section and the results section have been rewritten to clarify the study area and results. Please see point 6.
Point 3: Conceptually, I was not clear on whether the work focused on connectivity of the built environment, or soft mobility of the individual. This is an important distinction. Connectivity has to do with how environments can constrain or enable a particular behaviour, while soft mobility has to do with the individual (and sometimes group) decision about how to move around. Throughout the manuscript, the authors refer to "connectivity for soft mobility" and I was left wondering, exactly, what they meant by that term.
Response 3: The introduction has been amended to provide a short discussion of and definition for the phase "connectivity for soft mobility".
Point 4: Over the past 5 years there has been a growing interest (again) in Winter Cities. I encourage the authors to access this newer work, some of which is referenced or available at WinterCities.com.
Response 4: WinterCities.com is a great resource that I am familiar with and use. As a robust literature review on winter city design was published in 2018 in the Arctic Yearbook, I have referenced readers back to this publication rather than duplicating content.
Point 5: The authors also refer to participants' use of green and blue spaces. I don't know if it is directly applicable, but the authors might consider looking at Robin Kearns new book: Blue Space, Health and Wellbeing.
Response 5: Thank you for drawing my attention to this publication. I have looked at this book and whilst very interesting, I have not referenced it in this article. I have however added it to another article that we are in the process of writing.
Point 6: ON page 5 (lines 167-175) the findings here (no winter connectivity/mobility) seem to contradict the findings on page 4 (lines 148-161) that suggest that the ice road increases connectivity. I suspect that this is related to my first point - above - that there is some muddiness in the spatial extent of the two data collection methods.
Response 6: This comment is very helpful. Parts of the material and methods section and the results section including the former lines 148-161 have been rewritten to address this comment. Please also see point 2.
Point 7: The conclusions do seem somewhat self-evident. I, too, live in a Winter City. Certainly different planning strategies are required when considering how spaces will (or will not) be used in the different seasons. Could the authors, perhaps, provide more concrete examples of such proposed "re-tooling"? What would that look like? Because the reality of making such changes happen is complex. For example, cycling lanes or paths could be instituted (capital or infrastructure costs) but use throughout all four seasons required operating budgets to clear those paths of snow and ice. So such budgeting considerations must be part of any such 're-tooling'.
Response 7: To address this point concrete examples of new approaches to re-tooling planning strategies/ design solutions have been added. An example of extending blue/green planning strategies to blue/ green/ white planning strategies is now discussed in the conclusion. An illustration of new winter cycle lanes for the city of Kiruna (that are defined by projected light and snow) is also included as an example of design outcomes from such strategies.
Point 8: First, I found the manuscript overall far too brief. I am not sure if this is due to the journal's word limit, or simply the assumption of the authors that the readers are from Sweden and understand the constraints and challenges of that particular milieu. I would like to see the findings and discussion sections expanded and more complexity and nuance provided.
Response 8: The paper has been extensively rewritten to expand the complexity of the findings and discussion. This has resulted in the paper becoming circa 1000 words longer. Most of this additional text is added to the latter parts of the document.
Point 9: Second, the manuscript needs a careful and extensive edit for english language. For example, even the title states "...enable all year around...". This is not the conventional use in English. It is generally referred to "all year" or (more casually) "year round". Throughout the manuscript there are incomplete sentences ("That is...") extensive use of passive voice, and some awkward phrasing throughout. A simple pass through an English-speaking editor would easily catch and fix these simple errors.
Response 9: These English language errors have been addressed. The text was subject to English language proof reading. However, the document has been edited to improve English.

Reviewer 2 Report
This paper investigates a sub-arctic city in Sweden in order to understand how winter conditions affect connectivity for soft mobility. I have major reservations with the study- both in terms of framework and quality.
1. The study does not hold the quality of being a research, whereby there is a sound application of some methodology (qualitative or quantitative). Although the authors mentioned using mental mapping and photo-elicitation, it is not clear what is achieved by these. The findings (2 mental mapping illustrations and photos by participants) could have been readily achieved without any research- these do not provide any additional information to what is already known. The results are not based on systematic observations or in-depth research; hence, the study lacks in scientific soundness. In addition, there is not a clear research question or hypothesis.
2. Another major concern I have is regarding the definition of connectivity. A clear definition of connectivity and how it is measured is lacking, which leads to ambiguity.
3. The recruitment of participants are not clear. Why 15? Who are these people, how are they recruited?
4. It is not clear what is the contribution of this study is to the theoretical argument or practice.
5. The limitations of the study and suggestions for further work have not been discussed.
Some minor concerns are:
1. There are many English problems in the text, such as: (line 39 -a relatively little impact...); (lines 51-52 -are like others, are being challenged...); (line 82 -methods provides) Hence, the text should be revised.
2. The abstract is too repetitive in its use of wording (barriers and enablers to connectivity for soft mobility), and should be re-written to include more of the findings and what they mean in terms of both theory (contributions) and policy (practical implications).
3. The Introduction should be improved by providing a brief summary of the findings of earlier studies regarding connectivity for soft mobility. As it is, it is too limited to give a background to previous findings and arguments within this area. Hence, the contributions of this study is unclear.
Author Response
Point 1: The study does not hold the quality of being a research, whereby there is a sound application of some methodology (qualitative or quantitative). Although the authors mentioned using mental mapping and photo-elicitation, it is not clear what is achieved by these. The findings (2 mental mapping illustrations and photos by participants) could have been readily achieved without any research- these do not provide any additional information to what is already known. The results are not based on systematic observations or in-depth research; hence, the study lacks in scientific soundness. In addition, there is not a clear research question or hypothesis.
Response 1: Major parts of the materials and methods section have been rewritten to show a sound application of the methodology and systematic review of the results. Two research questions have been added. The conclusion has been extended to establish the implications of this research.
Point 2: Another major concern I have is regarding the definition of connectivity. A clear definition of connectivity and how it is measured is lacking, which leads to ambiguity.
Response 2: A definition of connectivity has been added to the introduction.
Point 3: The recruitment of participants are not clear. Why 15? Who are these people, how are they recruited?
Response 3: The materials and methods section has been rewritten clarify the recruitment process and number.
Point 4: It is not clear what is the contribution of this study is to the theoretical argument or practice.
Response 4: A practical implications and suggestions for further research section has been added. This and the extended conclusion addresses the research contribution of this paper.
Point 5: The limitations of the study and suggestions for further work have not been discussed.
Response 5: A limitations of this study section has been added. Further work is covered in the new section, practical implications and suggestions for further research.
Point 6: There are many English problems in the text, such as: (line 39 -a relatively little impact...); (lines 51-52 -are like others, are being challenged...); (line 82 -methods provides) Hence, the text should be revised.
Response 6: The document has been edited to improve English.
Point 7: The abstract is too repetitive in its use of wording (barriers and enablers to connectivity for soft mobility), and should be re-written to include more of the findings and what they mean in terms of both theory (contributions) and policy (practical implications).
Response 7: The abstract has been rewritten.
Point 8: The Introduction should be improved by providing a brief summary of the findings of earlier studies regarding connectivity for soft mobility. As it is, it is too limited to give a background to previous findings and arguments within this area. Hence, the contributions of this study is unclear.
Response 8: The introduction has been rewritten to highlight key previous findings. Key references for connectivity and winter urban design have also been added.

Reviewer 3 Report
The background material is generally well covered. While I agree that winter mobility is an area that is underresearched, there is some other research on winter mobility could be added. One example is:
Philippa Clarke, Jana A. Hirsch, Robert Melendez, Meghan Winters, Joanie Sims Gould, Maureen Ashe, Sarah Furst, and Heather McKay. Snow and Rain Modify Neighborhood Walkability for Older Adults. Can J Aging. 2017 Jun; 36(2): 159–169.
The background is also pretty well written. However, there are a few places where the grammar and clarity could be improved. I have marked some of them within the text.
Methodology
The methodology was very good and appropriate for this stage of development of the research area. I like the idea of mental mapping and development of plan based maps bringing disparate maps together. It might be good to note whether or not this map was reviewed by the participants before it was finalized.
The photo-elicitation method is quite appropriate and one that I have found to be very useful in getting people’s stories and setting context. The images are also very useful for talking with decision makers. I also like the fact that they encouraged participants to rank the photos in order from the most important barrier to the most important enabler.
However, the authors should also discuss whether the audio tape from this session was transcribed and how the ideas were analysed.
Results
The results were presented clearly and compellingly.
Discussion
The discussion was good and did not seem to go beyond the data. However, it would be important to link their work to findings and ideas from some of the work that was cited in the background. This would be ideally done at the top of Page 10 after the first sentence.

Author Response
Point 1: The background material is generally well covered. While I agree that winter mobility is an area that is underresearched, there is some other research on winter mobility could be added. One example is: Philippa Clarke, Jana A. Hirsch, Robert Melendez, Meghan Winters, Joanie Sims Gould, Maureen Ashe, Sarah Furst, and Heather McKay. Snow and Rain Modify Neighborhood Walkability for Older Adults. Can J Aging. 2017 Jun; 36(2): 159–169.
Response 1: This reference and others have been added.
Point 2: The methodology was very good and appropriate for this stage of development of the research area. I like the idea of mental mapping and development of plan based maps bringing disparate maps together. It might be good to note whether or not this map was reviewed by the participants before it was finalized.
Response 2: The materials and methods section has been rewritten to clarify all steps taken.
Point 3: The photo-elicitation method is quite appropriate and one that I have found to be very useful in getting people’s stories and setting context. The images are also very useful for talking with decision makers. I also like the fact that they encouraged participants to rank the photos in order from the most important barrier to the most important enabler. However, the authors should also discuss whether the audio tape from this session was transcribed and how the ideas were analysed.
Response 3: The materials and methods section has been rewritten to clarify all steps taken.
Point 4: The discussion was good and did not seem to go beyond the data. However, it would be important to link their work to findings and ideas from some of the work that was cited in the background. This would be ideally done at the top of Page 10 after the first sentence.
Response 4: The discussion and conclusion have been rewritten to link back to the introduction and key reasons for the research.

Round 2
Reviewer 2 Report
The authors have revised the manuscript effectively according to my comments.